# Glucose dysregulation in hospitalized non-critically ill patients with a suspected infection: A prospective study using continuous glucose monitoring

Anna D. Schoonhoven[1], Julia J. Bakker[2]*, Alessandra D. Di Mauro[2,3], Amarens van der Vaart[2,4], Riemer Been[2], André P. van Beek[2], Pratik Choudhary[5], Hjalmar R. Bouma[6,7], Peter R. van Dijk[8]

1 Emergency Department, University Medical Center Groningen, University of Groningen, Groningen, The Netherlands, 2 Department of Endocrinology, University Medical Center Groningen, Groningen, the Netherlands, 3 University of Milan, Milan, Italy, 4 Department of Nephrology, University Medical Center Groningen, University of Groningen, Groningen, the Netherlands, 5 University of Leicester Diabetes Research Centre Leicester, Leicester, United Kingdom, 6 Department of Internal Medicine, University Medical Center Groningen, University of Groningen, Groningen, The Netherlands, 7 Department of Clinical Pharmacy and Pharmacology, University Medical Center Groningen, University of Groningen, Groningen, The Netherlands, 8 Diabetes Center, Isala, Zwolle, The Netherlands

These authors contributed equally to this work.
* j.j.bakker@isala.nl

## Abstract

### Introduction

Dysglycaemia, defined as hypo- or hyperglycaemia, can occur during infection and is associated with worse outcomes during hospitalization. Previous studies on dysglycaemia in non-critically ill patients on general wards used point-of-care (POC) capillary measurements, possibly underestimating the problem. We assessed the prevalence and course of dysglycaemia in this population using continuous glucose monitoring (CGM).

### Methods

In this prospective, observational study at the University Medical Center Groningen, adults admitted to the Emergency Department with suspected infections were enrolled via the Acutelines data and biobank. Participants wore blinded CGM sensors (FreeStyle Libre Pro iQ) while continuing usual care. Episodes of dysglycaemia were defined as ≥15 minutes of glucose <3.9 mmol/L or >10 mmol/L. Primary outcome was the number of dysglycaemic episodes; secondary outcomes included duration, glucose levels, and associations with clinical outcomes.

### Results

CGM data from 90 participants (27% with a history of diabetes and 73% without) over a median of 3.4 days revealed 181 hyperglycaemia and 303 hypoglycaemia

**Data availability statement:** All relevant data are within the manuscript and its Supporting information files. Additional de-identified data from this study are available upon reasonable request from the corresponding author (j.j.bakker@isala.nl). Data access is subject to approval by the Medical Ethics Committee of the University Medical Center Groningen due to patient privacy restrictions.

**Funding:** P.C. and P.R.v.D. are funded by an unrestricted educational grant from the European Foundation for the Study of Diabetes (EFSD) mentorship program supported by AstraZeneca. Continuous glucose monitoring devices were provided free of charge by Abbott Diabetes Care. The establishment of Acutelines was funded by the University Medical Center Groningen. The funders had no role in study design, data collection and analysis, decision to publish, or preparation of the manuscript. https://www.europeandiabetesfoundation.org/recipients/future-leaders-mentorship-programme/.

**Competing interests:** The authors declare that the funders had no role in the study design, data collection, data analysis, interpretation, or writing of the manuscript. The authors declare no other competing interests.

episodes. In patients with a history of diabetes, 75% experienced hyperglycaemia (median of 6.5 events/patient). In contrast, 33% of individuals without prior diabetes experienced hyperglycaemia (median 1.5 events/ patient). Median Time in Range (glucose 3.9–10.0 mmol/L) was 59% for patients with and 86% for patients without known diabetes. Exploratory analyses showed no significant association between dysglycaemia and ICU admission or 30-day mortality.

## Conclusions

This observational study provides relevant insight into dysglycaemia among non-critically ill patients admitted to the hospital. Significant hyperglycaemia was observed in both participants with and without known diabetes. Therefore, CGM may enable earlier detection of dysglycaemia and thereby inform future interventional research and in-hospital strategies.

---

## Introduction

Approximately 30% of hospital admissions to general wards are related to infections [1]. In these patients, there is a high frequency (up to 40%) of both hypo- and hyperglycaemia ('dysglycaemia'). This can occur on those with known diabetes mellitus (DM), as well as those without a history of DM. Hyperglycaemia occurs because of increased production of pro-inflammatory cytokines and higher levels of hormones, such as cortisol, glucagon, and catecholamines [2].These changes lead to hepatic and skeletal glycogenolysis and gluconeogenesis, lipolysis in adipose tissue and also increased insulin resistance [3]. Additionally, hypoglycaemia can occur due to increased non-insulin-mediated glucose uptake in tissues rich in macrophages, such as in the spleen, ileum, liver and lungs [4] and due to reduction in food intake seen with illness. This chain of events negatively impacts both the innate and adaptive immune response, which leads to decreased bacterial clearance, and subsequently a longer and more profound course of the infection [5]. Together, these processes reflect the acute metabolic stress and systemic inflammatory response induced by infection, which may provoke early disturbances in glucose homeostasis even in individuals without diabetes.

Inpatient hypo-and hyperglycaemia, in participants with or without a prior diagnosis of DM, is associated with an increased risk of complications and mortality, longer hospital stay, higher admission rate to the intensive care unit (ICU), and higher need for intensive support and disability after hospital discharge [6–12]. While improved identification and timely management may mitigate harm, evidence in non-ICU patients remains limited.

Current standard care relies on point-of-care (POC) testing of capillary blood glucose concentrations via finger pricks 2–4 times a day, often pre-meal. Since this method only provides snapshots of glucose concentrations, it results in missed episodes of dysglycaemia, especially those after meals, estimated to be around 30% [13,14]. Continuous glucose monitoring (CGM) provides a solution to this challenge,

as it offers real-time measurements that are – depending upon the specific type of sensor – performed every 5–15 minutes in the interstitial fluid [15,16]. As such, CGM enables earlier and more detailed insights in glucose levels and trends. Over the past five years, studies on general wards comparing CGM to POC have shown that CGM detects more episodes of hypo- and hyperglycaemia and is associated with reduced glycemic variability and time spent in hyperglycaemia [16–22]. Most of these previous studies, however, were performed in mixed populations. Given the large size of the population admitted to the hospital with an infection, the negative impact of dysglycaemia on outcomes, and the potential advantages of CGM over POCT, we aimed to extend current knowledge on the use of CGM among inpatients. Therefore we prospectively investigated dysglycaemia measured using CGM amongst participants hospitalized with a suspected infection.

## Materials and methods

The writing of this paper was guided by The Strengthening the Reporting of Observational Studies in Epidemiology (STROBE) Statement criteria [23].

### Design and participants

In this prospective, observational study, adult patients, with and without known DM, who were consecutively admitted to the Emergency Department (ED) between July 2022 and July 2023, and who were included in the Acutelines databiobank in the University Medical Center Groningen (UMCG), The Netherlands, were enrolled in the study. Acutelines is a multi-disciplinary prospective hospital-based cohort study recording the complete acute patient journey of people admitted to the ED of the UMCG [24]. Participants were asked for written informed consent, by proxy when necessary. The Acutelines cohort study is approved by the medical ethics committee of the UMCG and registered under trial registration number NCT04615065 at ClinicalTrials.gov. The study was conducted in accordance with the Declaration of Helsinki.

For the present study, we included all adult patients (≥18 years) admitted through the emergency department with a clinical suspicion of infection, as determined by routine ED practice. Suspicion could arise from clinical judgement, abnormal vital signs (e.g., ≥ 2 SIRS or ≥2 qSOFA criteria), or infection-related triggers such as blood cultures drawn or abnormal temperature, particularly in high-acuity patients. This reflects real-life emergency department practice, where triage and treatment decisions rely on clinical suspicion of infection without microbiological confirmation.

To ensure accuracy of interstitial glucose measurements with the CGM, we excluded patients with systolic blood pressure < 100 mmHg within 60 minutes of ED admission, without the need of vasopressor medication. In addition, we excluded participants who used glucocorticosteroids, as these drugs have substantial and variable effects on glucose metabolism that could confound primary outcome. Participants who received corticosteroids after enrolment were included up to the point of administration.

Fig 1 provides a diagram of the participants during the study period: of the total of 143 patients were enrolled, 53 received corticosteroids and were excluded. The final analysis was based on 90 patients.

### Procedures

All enrolled participants were provided with a blinded CGM (FreeStyle Libre Pro iQ, Abbott [25]), on the back of the upper arm. This sensor measures interstitial glucose at 15-minute intervals. Throughout the study, both participants and hospital staff were blinded to the outcomes of the CGM measurements, and treatment decisions were not influenced by these readings. CGM was conducted for a maximum of 7 days, after which the sensor was removed, and data were downloaded. The sensor was removed pre-emptively if a patient required magnetic resonance imaging, computed tomography scans, or high-frequency electrical heat (diathermy). Additionally, the sensor was removed upon admission to the Intensive Care Unit. Glucose management, for patients with and without a prior history of DM, was performed according to the national protocol for in-hospital glucose management [26]. This protocol aims to maintain glucose levels between 4 and 10 mmol/L and promotes the use of sliding scale (correction) doses of short acting insulin aspart, irrespective of meals.

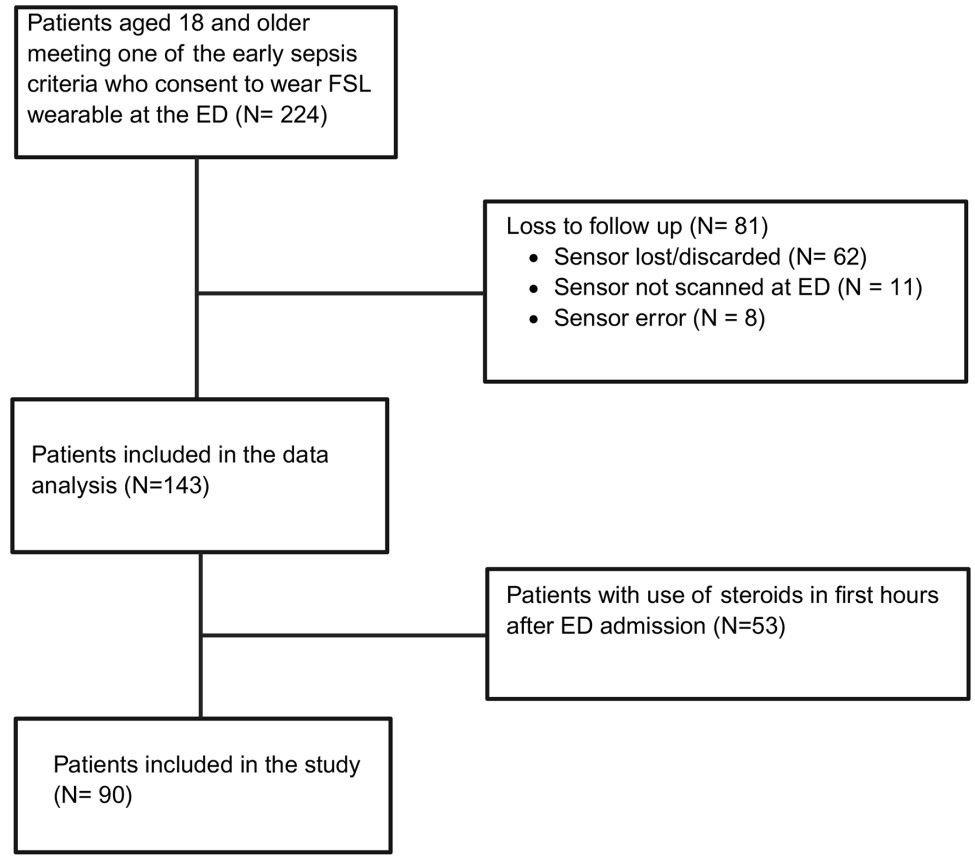

**Fig 1. Participant flowchart.**

For blood glucose levels between 10 and 15 mmol/L, a subcutaneous bolus of two units of insulin was administered. For glucose values between 15 and 20 mmol/L, four units of insulin were administered and six units of insulin were administered for glucose values >20 mmol/L.

## Outcome measures

Primary outcome was the number of dysglycaemic (both hypo- and hyperglycaemic) episodes measured with CGM among all participants. An episode was defined as a period of at least 15 minutes with either hypo- or hyperglycaemia [27]. Secondary outcomes included the number, duration and height of hypo- and hyperglycaemic episodes. Comparisons were made between participants with and without a previous history of DM. As exploratory secondary outcomes, we investigated the association of glucose metrics with adverse outcomes including a composite of ICU admission and 30-day mortality and length of hospital stay.

Hypoglycaemia was defined as a glucose value <3.9 mmol/L for at least 15 minutes, while hyperglycaemia was defined as a glucose level >10.0 mmol/L for at least 15 minutes [28–30]. Furthermore, time in range (TIR) is defined as per ATTD consensus statement as percentage of time spent between 3.9–10 mmol/L, time in tight range (TITR) as percentage of time spent between 3.9–7.8 mmol/L, time below range (TBR) as percentage of time <3.9 mmol/L, time above range (TAR) as percentage of time >10.0 mmol/L [31].

## Statistical analysis

A MATLAB (R2023 Academic Use; MathWorks, Natick, MA, USA) algorithm extracted CGM features. Statistical analysis was performed using RStudio 2023.06.1 Build 524 (RStudio Team. RStudio: Integrated Development for R. RStudio, Inc., Boston, MA, USA). Normality of variables was assessed using histograms and QQ-plots. Baseline data were compared using the Chi-square test for categorical variables and Student's t-test or Mann-Whitney U test for normally distributed or skewed continuous data, respectively. Correlations between CGM metrics were analysed using Pearson's r or Spearman's rho. A p-value <0.05 was considered statistically significant. Given the absence of missing data related to the outcomes, additional missing data analyses were not necessary. Data analysis was limited to the first 120 hours (5 days), as there were insufficient participants with complete data for the full 7-day period, primarily due to early sensor removals for medical reasons. Given the exploratory nature of this study and the limited sample size, no formal adjustment for multiple comparisons was applied [32]. Regression analyses to adjust for confounders such as age or BMI were considered but not performed due to the small number of adverse clinical events [33]. Instead, exploratory univariate regression analyses were conducted to assess associations between dysglycaemia and clinical outcomes.

## Results

Baseline characteristics for all 90 participants are shown in Table 1. The median age was 67 years (IQR [54–73]), with 58% men and body mass index (BMI) of 25.1 [22.1–30.6] kg/m². Among them, 24 participants (27%) had a prior history of type 2 DM. All of these participants used glucose-lowering medication, including insulin (62%), metformin (46%), gliclazide (25%), DPP4 inhibitors (4%) or SGLT2 inhibitors (12%). Compared to those without known DM, participants with DM were older (70 vs. 64 years old, p = 0.017), more likely to be smokers (87% vs. 53%, p = 0.007), had a higher BMI (30.6 vs. 24.2, p < 0.001), a higher Charlson Comorbidity Index (5 vs 3, p < 0.013), and presented with higher values at ED arrival for plasma glucose (9.7 vs. 6.8 mmol/L, p < 0.001), C-reactive protein (141.0 vs. 60.5 mg/L, p = 0.014) and creatinine (128.0 vs. 80.0 µmol/L, p = 0.001).

The median study period of all participants was 82 hours (IQR [41–114]). CGM measured dysglycaemia was present in 73 participants (81%) and occurred throughout the admission (Fig 2). As shown in Table 2, there were 181 hyperglycaemic events in 40 participants (44%) with a median of 3 events per person (IQR [1–7]). During these events, the median glucose concentration was 10.7 mmol/L (IQR [10.4, 11.3]) with a maximum concentration of 22.8 mmol/L and a median duration of 1 hour and 15 minutes (IQR [0h40m, 4h38m]). We observed 303 hypoglycaemic episodes, in 50 participants (56%), with a median of 5 episodes per person (IQR [2,9]). During these events, the median glucose concentration was 3.7 mmol/L (IQR [3.5, 3.7]), with a minimum concentration of 2.2 mmol/L and a median duration of 58 minutes (IQR [0h37m, 1h41m]). As shown in Table 3, the median glucose level among all participants throughout hospitalization, as measured with CGM, was 6.0 mmol/L (IQR [5.2–6.9]) with a coefficient of variation (CV) of 19.4% (IQR [14.8–24.4]). TIR, TITR, TBR and TAR were 93.2% (IQR [76.4–99.0]), 90.4% (IQR [72.3–99.1]), 1.7% (IQR [0.0–8.8]) and 0% (IQR [0.0–4.6] %), respectively.

In the group of participants with a history of DM, 18 participants (75%) experienced 119 hyperglycaemic events with glucose levels above 10.0 mmol/L (Table 2). In contrast, 22 participants without a history of DM (33%) experienced 62 hyperglycaemic events. Hyperglycaemic events in participants with DM had a higher median glucose level and lasted longer compared to those without DM: median glucose levels of 11.5 mmol/L (IQR [11.1, 12.1] vs. 10.4 mmol/L (IQR [10.3, 10.6]) and median event durations of 5 hours 15 minutes (IQR [02h09m, 7h01m]) vs. 44 minutes (IQR [00h25m, 01h00m]). See also Fig 3 where a standardized episode of hyperglycaemia > 10.0 mmol/L for at least 15 minutes is presented for both participants with and without a history of DM. Participants with a history of DM more often experienced CGM measured episodes of glucose > 13.9 mmol/L. As presented in Fig 2 participants with a history of DM experienced dysglycaemia throughout admission while those without a history of DM primarily had episodes during the first 72 hours of admission.

**Table 1. Baseline characteristics of study population.**

| | All N = 90 | Without DM N = 66 | With DM N = 24 | P value |
|---|---|---|---|---|
| **Demographics** | | | | |
| Age | 67 [54, 73] | 64 [49, 73] | 70 [66, 76] | 0.017* |
| Men (%) | 52 (58) | 35 (53) | 17 (70) | 0.204 |
| BMI [kg/m²] | 25.1 [22.1, 30.6] | 24.2 [21.3, 27.0] | 30.6 [24.7, 35.1] | <0.001* |
| Smoker (%) | 55 (62) | 34 (53) | 21 (87) | 0.007* |
| Alcohol use (%) | 24 (27) | 19 (30) | 5 (21) | 0.574 |
| CCI-score | 4 [2, 6] | 3 [2, 6] | 5 [4, 6] | 0.013* |
| **Vital Parameters** | | | | |
| Systolic blood pressure [mmHg] | 130 [113, 148] | 130 [112, 148] | 126 [115, 147] | 0.890 |
| Diastolic blood pressure [mmHg] | 75 [69, 87] | 75 [69, 90] | 75 [66, 83] | 0.725 |
| Heart rate [bpm] | 102 [91, 110] | 102 [92, 110] | 97 [83, 111] | 0.404 |
| Respiratory rate [bpm] | 20 [16, 23] | 20 [16, 23] | 22 [18, 23] | 0.277 |
| SpO₂ [%] | 96 [94, 98] | 96 [94, 98] | 95 [92, 98] | 0.451 |
| Temperature [°C] | 37.5 [36.8, 38.4] | 37.6 [36.8, 38.4] | 37.5 [36.6, 38.2] | 0.707 |
| **Disease severity scoring** | | | | |
| qSOFA | 0 [0, 1] | 0 [0, 1] | 0 [0, 1] | 0.365 |
| SIRS | 2 [1, 3] | 2 [1, 3] | 2 [2, 3] | 0.947 |
| **Laboratory values** | | | | |
| Glucose mmol/L | 7.2 [6.2, 9.2] | 6.8 [6.0, 7.5] | 9.7 [7.8, 13.5] | <0.001* |
| CRP [mg/L] | 69.5 [23.5, 149.7] | 60.5 [21.5, 124.0] | 141.0 [44.2, 294.0] | 0.014* |
| WBC [10⁹/L] | 11.5 [6.8, 15.4] | 11.0 [5.8, 15.0] | 12.3 [8.2, 16.5] | 0.349 |
| Creatinine [µmol/L] | 90 [67, 124] | 80.0 [64.2, 114.7] | 128.0 [88.5, 176.0] | 0.001* |
| | | | | |

General descriptive characteristics of our population and comparison between participants with previous history of diabetes and not. Data are presented as median [25th, 75th percentile]. BMI: body mass index, CCI: Charlson Comorbidity Index, CRP: C reactive protein, WBC: white blood cells. Significant p values are marked *.

Throughout hospital admission participants with a history of DM also had a significantly higher mean glucose level (8.5 mmol/L (IQR [6.3–10.8]) vs. 5.7 mmol/L (IQR [5.0–6.3]), p < 0.0001) and a significantly lower TIR (59% (IQR [38–86]) vs. 96% (IQR [88–100]), p < 0.0001) and TITR (40% (IQR [14–73]) vs. 96% (IQR [85–100]), p < 0.0001), compared to patients without DM (Table 3). In participants without a history of DM there were 254 episodes of hypoglycaemia with a median duration of 51 minutes (IQR [0h36m, 1h36m]) while participants with a history of DM had 49 hypoglycaemic episodes with a median of 1 hour 27 minutes (IQR [00h55m, 02h30m]). Throughout admission there was no significant difference between the groups in the time in the hypoglycaemic range (Table 3).

A total of 5 participants were admitted to the ICU during the hospital stay and 6 participants died within 30 days after admission. Univariate linear regression showed that participants with dysglycaemia had a non-significant increase in length of hospital stay of 1.33 days compared with those without dysglycaemia (β = 1.33, 95% CI –1.20 to 3.85, p = 0.30). Univariate logistic regression indicated that dysglycaemia was not significantly associated with the composite outcome of 30-day mortality or ICU admission (OR = 0.42, 95% CI 0.10–2.16, p = 0.25).

## Discussion

The present study investigated dysglycaemia among inpatients admitted with a suspected infection. Overall, the observational data demonstrate that CGM identified dysglycaemia in most (81%) non-critically ill participants admitted to general

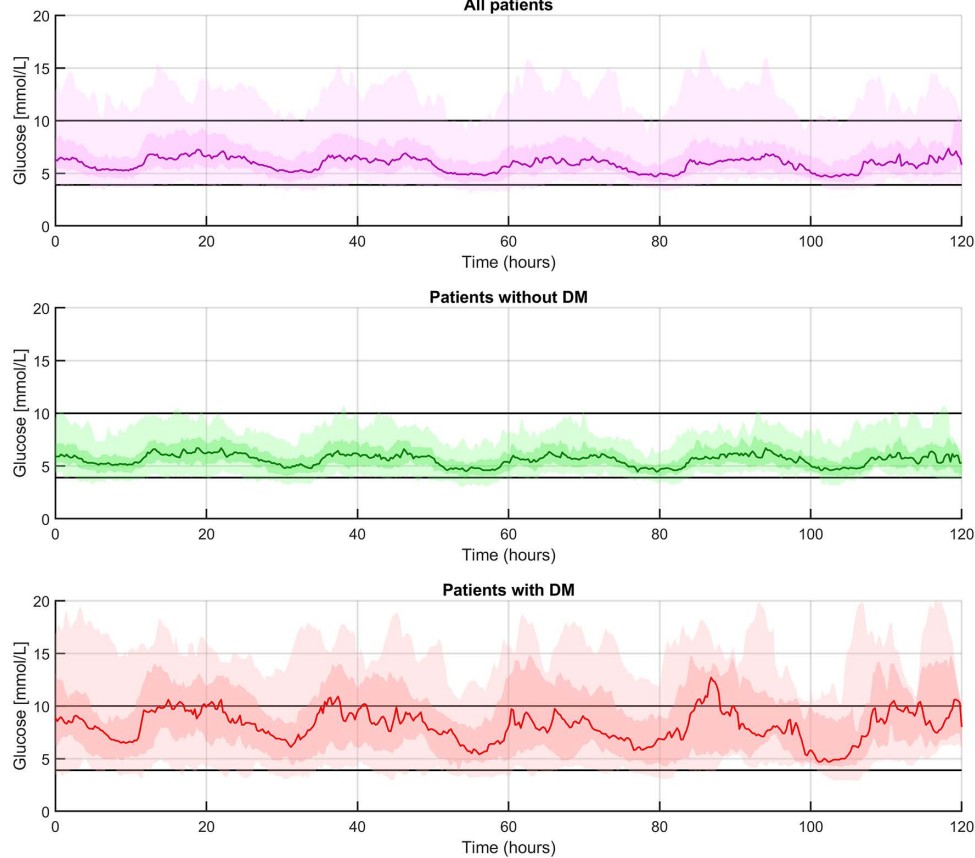

**Fig 2. Glucose profiles during the first 5 days after ED admission.** The solid colored line represents the median glucose level, the darker shaded area indicates the interquartile range (25th to 75th percentile), and the lighter shaded area represents the 5th to 95th percentile. The black horizontal lines indicate the target glucose range.

wards of the hospital with an infection. Although the sample size in this observational study was modest and confounders should be taken into consideration, the use of blinded CGM in this population led to several other observations, i.e., hyperglycaemia (glucose >10.0 mmol/L) occurred in 44% of participants, predominantly in those with pre-existing DM, with episodes lasting nearly 5 hours. Hypoglycaemia affected 55% of participants. Time in the euglycaemic range (3.9–7.8 mmol/L) was low in patients with diabetes: median 40% [IQR 14–73]. In participants without DM, hyperglycaemia was mostly seen during the first 72 hours, whereas in participants with DM it persisted throughout admission. These findings highlight the high prevalence and persistence of dysglycaemia in infected hospitalized patients. The observed differences in hypo- and hyperglycaemia patterns between patients with and without DM are likely related to physiological and treatment-related factors such as persistent insulin resistance in DM or stress-induced dysregulation in participants without DM.

Given the strong association of hyperglycaemia with adverse outcomes of hospitalization for an infection [6–12], the findings of this observational study emphasizes the necessity of closely monitoring glucose levels in all hospitalized patients with an infection. Although evidence suggests that earlier detection and subsequent earlier treatment of dysglycaemia results in improved glycaemic control [20], it remains to be tested if this also leads to improved outcomes of hospitalization. The clinical significance of hyperglycaemia during infection remains debated: some studies suggest that hyperglycaemia worsens outcomes, while others argue that it is merely an epiphenomenon [6,7]. The data from our

**Table 2. Details of CGM measured episodes of hypo- and hyperglycaemia.**

| | | All N = 90 | Without DM N = 66 | With DM N = 24 |
|---|---|---|---|---|
| **< 3.0 mmol/L** | Total episodes | 30 | 19 | 11 |
| | Participants with episodes (%) | 17 (18.9) | 14 (21.1) | 3 (12.5) |
| | Number of episodes | 1 [1,2] | 1 [1, 2] | 5 [2,5] |
| | Duration of episodes | 00h31m [00h22m, 01h07m] | 00h26m [00h21m, 00h37m] | 02h34m [02h22m, 03h01m] |
| | Average glucose | 2.9 [2.9, 2.9] | 2.9 [2.9, 2.9] | 2.9 [2.8, 2.9] |
| **< 3.9 mmol/L** | Total episodes | 303 | 254 | 49 |
| | Participants with episodes | 50 (55.5) | 40 (60.6) | 10 (41.6) |
| | Number of episodes | 5 [2, 9] | 6 [2, 9] | 5 [2, 6] |
| | Duration of episodes | 00h58m [00h37m, 01h41m] | 00h51m [00h36m, 01h36m] | 01h27m [00h55m, 02h30m] |
| | Average glucose | 3.7 [3.5, 3.7] | 3.7 [3.6, 3.7] | 3.6 [3.5, 3.7] |
| **> 7.8 mmol/L** | Total episodes | 417 | 296 | 121 |
| | Participants with episodes | 64 (71.1) | 42 (63.6) | 22 (91.7) |
| | Number of episodes | 5 [3, 9] | 5 [3, 9] | 5 [3, 8] |
| | Duration of episodes | 01h32m [00h45m, 04h29m] | 01h07m [00h38m, 01h45m] | 08h01m [02h39m, 11h15m] |
| | Average glucose | 8.5 [8.3, 9.1] | 8.4 [8.3, 8.7] | 9.7 [8.8, 11.0] |
| **> 10.0 mmol/L** | Total episodes | 181 | 62 | 119 |
| | Participants with episodes | 40 (44.4) | 22 (33.3) | 18 (75.0) |
| | Number of episodes | 3 [1, 7] | 2 [1, 3] | 7 [3, 9] |
| | Duration of episodes | 01h15m [00h40m, 04h38m] | 00h44m [00h25m, 01h00m] | 05h15m [02h09m, 07h01m] |
| | Average glucose | 10.7 [10.4, 11.3] | 10.4 [10.3, 10.6] | 11.5 [11.1, 12.1] |
| **> 13.9 mmol/L** | Total episodes | 74 | 1 | 73 |
| | Participants with episodes | 15 (16.6) | 1 (1.5) | 14 (58.3) |
| | Number of episodes | 5 [1, 8] | 1 [NA]* | 6 [2, 8] |
| | Duration of episodes | 02h06m [01h04m, 03h10m] | 00h31min [NA]* | 01h48m [01h09m, 03h13min] |
| | Average glucose | 14.7 [14.4, 15.5] | 14.3 [NA]* | 15.0 [14.4, 15.6] |

Total episodes, number of patients with episodes (%), number of episodes per patient (median [25th, 75th percentile] among those with episodes), episode duration (median [25th, 75th percentile] among those with episodes), and average glucose levels during episodes (median [25th, 75th percentile]). *For groups with n = 1, only the median is reported, as the interquartile range (IQR) is not applicable.

current study do not resolve this debate, as the exploratory analysis of the relationship between dysglycaemia and clinical outcomes was limited by the low number of ICU-admission and deaths. This small number of events does not allow multivariable analyses to adjust for potential confounders as this would lead to overfitting. The exploratory findings may reflect early metabolic stress and could therefore function as a potential biomarker for clinical deterioration. Confirmation of the prognostic relevance of dysglycemia requires larger, adequately powered muticenter studies.

Previous studies towards in-hospital dysglycaemia have mainly been conducted in critically ill patients in an ICU setting, where—due to the severity of illness—hyperglycaemia is more pronounced than in our study. On the other hand, these studies primarily used POC glucose measurements, potentially missing episodes of hyperglycaemia. This could explain the (paradoxically) higher time in the euglycemic range found in other studies with more critically ill patients than in our population, such as the retrospective study of ICU patients where the median time-in-range in individuals with DM was 84.1% [34] as compared to 59.1% in our population. In our dataset, CGM detected more episodes of hyperglycaemia (> 10 mmol/L) than POC (263 vs 136) and CGM detected hyperglycaemia 3.5 hours [1.0, 4.4] earlier. These data align with recent non-ICU trials showing that CGM detects more dysglycemic events, 2–3 hours earlier, alongside higher time-in-range and less time-above-range with CGM-guided glucose management [16–20]. Although this was not the aim of our

**Table 3. Continuous glucose monitoring metrics.**

| | All N = 90 | Without DM N = 66 | With DM N = 24 | P value |
|---|---|---|---|---|
| Time of measurement (hours) | 82 [41, 114] | 66 [40, 118] | 92 [62, 111] | 0.500 |
| Mean glucose [mmol/L] | 6.0 [5.2, 6.9] | 5.7 [5.0, 6.3] | 8.5 [6.3, 10.8] | <0.001* |
| SD [mmol/L] | 1.2 [0.8, 1.6] | 1.0 [0.8, 1.3] | 2.3 [1.4, 3.0] | <0.001* |
| CV [%] | 19.4 [14.8, 24.4] | 18.1 [14.1, 21.7] | 26.7 [20.3, 33.0] | <0.001* |
| Time in range [%] | 93.2 [76.4, 99.0] | 96.0 [88.5, 99.5] | 59.1 [38.2, 85.5] | <0.001* |
| Time in tight range [%] | 90.4 [72.3, 99.1] | 95.7 [84.6, 99.7] | 39.5 [14.1, 72.8] | <0.001* |
| Time in Hypoglycaemia [%] | 1.7 [0.0, 8.8] | 2.0 [0.0, 10.7] | 0.0 [0.0, 3.9] | 0.069 |
| Time in Hyperglycaemia [%] | 0.0 [0.0, 4.6] | 0.0 [0.0, 0.8] | 31.6 [4.4, 60.2] | <0.001* |

Continuous glucose monitoring variables for the overall population and comparison between participants with and without a history of diabetes. Time in range: 3.9–10 mmol/L [70–180 mg/dL]: Time in tight range: 3.9–7.8 mmol/L [70–140 mg/dL], time in the hyperglycaemic range: > 10.0 mmol/L [>180 mg/dL]; time in hypoglycaemic range <3.9 mmol/L [<70 mg/dL]. Data are presented as median [25ᵗʰ, 75ᵗʰ percentile]. Significant p values are marked * Abbreviations: SD, standard deviation; CV, coefficient of variation.

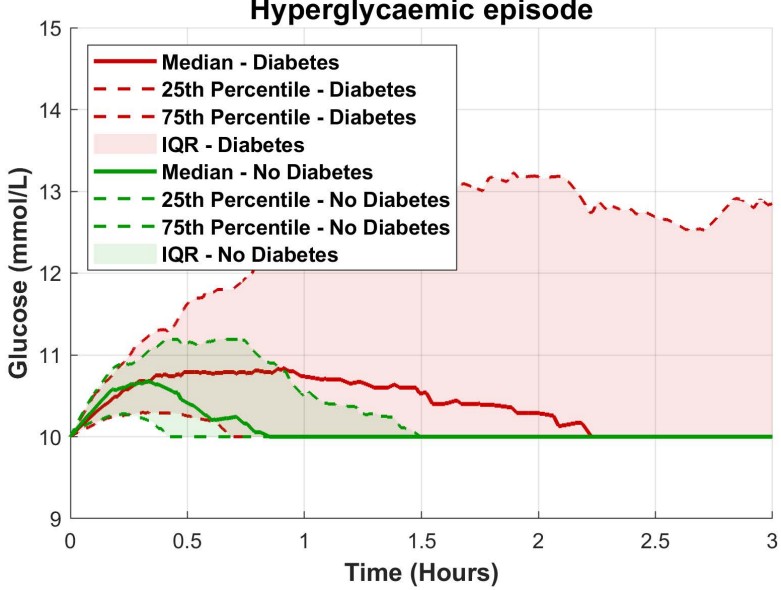

**Fig 3. Course of an average episode of hyperglycaemia in participants with and without diabetes.** Hyperglycaemia episodes (≥10.0 mmol/L for at least 15 minutes) were identified for both the diabetes and non-diabetes groups. Each episode was aligned to start at time t = 0, after which mean glucose values were calculated for each time point. To address variability in episode durations (and prevent disproportionately long episodes from skewing comparisons), episodes were standardized by assuming glucose levels remain at 10.0 mmol/L once they drop below this threshold.

study and these results need confirmation, it may emphasize the potential of CGM as compared to POC for early detection of in-hospital dysglycaemia.

We observed a high frequency of hypoglycaemia: 55% of all participants experienced these, with an even higher percentage in the group participants without a history of DM (61 vs. 42%). Taken the relative short duration and only slightly lowered median glucose levels of these hypoglycaemic episodes into account (58 minutes [37m, 1h41m] and 3.7 mmol/L [3.5, 3.7]), most of these episodes were level 1 hypoglycaemia and data suggest that these are common in up to 30%

of those without diabetes and represent normal ranges of glucose [35]. The blinded FreeStyle Libre Pro IQ has a known tendency to register more hypoglycaemia near the 3.9 mmol/L threshold (particularly in the first 24 hours and during rapid glucose change) which may overestimate level 1 hypoglycaemia [36]. To address this, we performed an additional analysis of hypoglycaemia distribution across sensor days. Although the absolute number of events was highest on day 1, the normalized rate of hypoglycaemia per patient remained relatively stable across days (2.8, 2.33, 2.50, 1.81 and 1.85 events for days 1–5, respectively), indicating that hypoglycaemia was not disproportionately concentrated during the first 24 hours. Given that first-day exclusion would be unlikely to alter the overall conclusions, and would simultaneously remove clinically relevant glucose dynamics, we retained these measurements. Overall, the hypoglycaemic events were brief and mild, and most likely represent physiological or sensor-related fluctuations rather than clinically significant hypoglycaemia.

The accuracy of the blinded CGM we used in this study could theoretically cause an over- or underestimation of our findings. In our study, strict exclusion criteria were applied, including exclusion of patients with hypotension (defined as systolic blood pressure <100 mmHg after 30 and 60 minutes from admission without vasopressor support) to mitigate confounding factors that could affect CGM performance. The CGM used in this study is a well-validated and studied device in the inpatient settings, with a mean absolute of relative difference (MARD) of 12.7 to 14.8% [16,37]. As a post-hoc analysis we analysed accuracy (MARD, ±15%/15 mg/dL, ±20%/20 mg/dL, and ±30%/30 mg/dL agreement rates and percentage of matched pairs in the Clarke Error Grid (CEG) Analysis zones) and the number of discrepancies and time difference of episode between CGM and POC (S1 Fig). Overall, the MARD in the present study (14.1%) is comparable to previous reports. Interestingly, although it should be stressed that the CGM device we used in this study was blinded and not suitable nor intended for clinical decision making, >98% of readings were in zone A and B (zones considered to be clinically acceptable, indicating that the observed bias from POC measurements would not lead to treatment decisions that could put the patient at risk) [38].

This observational study has several limitations. First, the missing data, primarily due to missing sensors, could impact our outcomes. During the study, unfamiliarity with CGM among hospital staff and participants often resulted in early removal or disposal of the sensor. This is an important observation and should be considered when potentially implementing CGM into the hospital setting. Also, patients that received steroids were excluded as steroids could affect glucose metabolism and thereby distort the outcomes. However, this resulted in the exclusion of a substantial number of participants (n = 53). Taken together, the sample size of this study is therefore limited, which, for example, also prevented us from performing multivariate analyses. Another limitation is the lack of detailed information on factors that may influence mechanisms underlying dysglycemia, such as nutritional intake, type of infection or the use of blood glucose-lowering medication. This limits interpretability and should be incorporated in future studies. Furthermore, we were unable to assess HbA1c levels in our participants, which could have provided insights into long-term glycemic control or even the presence of undiagnosed DM prior to admission. Finally, the limited sample size should be considered when interpreting the results. This led to a small number of adverse clinical events which led to limited statistical power to detect large effects. The absence of significant associations between dysglycemia and clinical outcomes should therefore be interpreted with caution [39].

## Conclusions

Among participants hospitalized with a suspected infection, CGM measured dysglycaemia was present in 81% of non-critically ill patients with an infection at any time during their hospital stay. Among those with a history of diabetes, 75% experienced prolonged and significant episodes of both hypo- and hyperglycaemia. Interestingly, dysglycaemia was also common in patients without a history of diabetes. These data call for more proactive and tailored glycaemic detection strategies for individuals. CGM may offer a viable tool for improving the detection of dysglycemia and could serve as a basis for future interventional studies aimed at optimizing in-hospital glucose management.

## Supporting information

**S1 Fig. Clarke Error Grid analysis.** Distribution of matched glucose pairs across Clarke Error Grid zones (A–E), comparing point-of-care glucose values with FreeStyle Libre measurements. The analysis shows 75% of values in Zone A (clinically accurate), 24% in Zone B (benign errors), 0% in Zone C (errors potentially leading to unnecessary treatment), and 1% in Zone D (errors indicating failure to detect hypo- or hyperglycaemia).
(TIFF)

**S1 Table. Detailed accuracy metrics of CGM vs. POC.** Summary of matched pairs, MARD values, and agreement percentages across different time intervals (0–24h, 24–48h, etc.) and glucose ranges (hypoglycemia, time in range, hyperglycaemia).
(DOCX)

## Acknowledgments

The authors thank all patients and staff contributing to Acutelines. During the preparation of this work the authors used ChatGPT-5 (OpenAI, San Francisco, CA, USA) to improve readability and refine the language. After using this tool, the authors reviewed and edited the content as needed and take full responsibility for the content of the publication.

## Author contributions

**Conceptualization:** Julia J. Bakker, André P. van Beek, Pratik Choudhary, Hjalmar R. Bouma, Peter R. van Dijk.

**Data curation:** Julia J. Bakker, Amarens van der Vaart, Riemer Been.

**Formal analysis:** Anna D. Schoonhoven, Julia J. Bakker, Alessandra D. Di Mauro.

**Funding acquisition:** Pratik Choudhary, Peter R. van Dijk.

**Investigation:** Anna D. Schoonhoven, Julia J. Bakker, Alessandra D. Di Mauro.

**Methodology:** Anna D. Schoonhoven, Julia J. Bakker, Alessandra D. Di Mauro, Hjalmar R. Bouma, Peter R. van Dijk.

**Resources:** Peter R. van Dijk.

**Software:** Anna D. Schoonhoven, Alessandra D. Di Mauro.

**Supervision:** Hjalmar R. Bouma, Peter R. van Dijk.

**Visualization:** Anna D. Schoonhoven, Alessandra D. Di Mauro.

**Writing – original draft:** Anna D. Schoonhoven, Alessandra D. Di Mauro, Peter R. van Dijk.

**Writing – review & editing:** Anna D. Schoonhoven, Julia J. Bakker, Alessandra D. Di Mauro, Amarens van der Vaart, Riemer Been, André P. van Beek, Pratik Choudhary, Hjalmar R. Bouma, Peter R. van Dijk.

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
