## [Decision Letter · Decision Letter 0]

23 Oct 2025

Dear Dr. Bakker,

Thank you for submitting your manuscript to PLOS ONE. After careful consideration, we feel that it has merit but does not fully meet PLOS ONE’s publication criteria as it currently stands. Therefore, we invite you to submit a revised version of the manuscript that addresses the points raised during the review process.

We look forward to receiving your revised manuscript.

Kind regards,

Hidetaka Hamasaki

Academic Editor

PLOS ONE

Additional Editor Comments (if provided):

Reviewers' comments:

Reviewer's Responses to Questions

**Comments to the Author**

1. Is the manuscript technically sound, and do the data support the conclusions?

Reviewer #1: Partly

Reviewer #2: Partly

2. Has the statistical analysis been performed appropriately and rigorously?

Reviewer #1: I Don't Know

Reviewer #2: Yes

3. Have the authors made all data underlying the findings in their manuscript fully available?

Reviewer #1: Yes

Reviewer #2: Yes

4. Is the manuscript presented in an intelligible fashion and written in standard English?

Reviewer #1: Yes

Reviewer #2: Yes

Reviewer #1: This manuscript by Schoonhoven et al., titled “Glucose dysregulation in hospitalized non-critically ill patients with a severe infection: a prospective study using continuous glucose monitoring” investigated the prevalence and course of dysglycaemia in non-critically ill adults hospitalized for severe infection using CGM, the authors find that in 90 patients, dysglycaemia is common in both diabetic and non-diabetic inpatients with infection and CGM could improve detection and management. The study provides interesting pilot data but the small sample size and low number of clinical events limit the robustness of conclusions, particularly regarding associations between dysglycaemia and outcomes. Larger multicenter studies are needed to confirm these findings. Additionally, several methodological and interpretative issues need to be addressed.

In the method section of study design, Inclusion and exclusion criteria are clear, but justification for excluding corticosteroid users should be elaborated, does this exclusion bias the sample toward less severe infections? Please clarify whether “severe infection” per inclusion criteria.

In Statistical Analysis, it is stated that p<0.05 was considered significant, but multiple comparisons were made without correction. Please comment on whether a correction (e.g., Bonferroni or FDR) was considered. Was any regression modeling performed to adjust for potential confounders such as age, BMI, or diabetes status when exploring associations with dysglycaemia?

In the discussion, the authors conclude that CGM may “allow earlier intervention” but as the study was blinded and observational, this inference is speculative. Please temper this statement or support it with cited intervention studies.

The reported 55% rate of hypoglycaemia (including in non-diabetic patients) seems high. Could sensor compression or first-day calibration bias explain part of this? Consider adding a sensitivity analysis excluding the first 24 hours of CGM data.

Reviewer #2: The topic is clinically relevant and timely, as glucose instability in acute infection is common but under-characterized outside the ICU setting. The prospective design and adherence to STROBE standards are strengths. However, the study as presented remains primarily descriptive and exploratory. The analyses are insufficiently detailed, the comparisons lack statistical rigor, and several methodological and interpretive limitations reduce the strength of the conclusions. Substantial revision is required before this manuscript can be considered for publication.

1. The study relies mainly on descriptive and univariate comparisons between patients with and without diabetes. To enhance interpretability, apply multivariate regression (logistic or linear, as appropriate) to evaluate associations between dysglycaemia metrics (such as mean glucose, time in range) and outcomes (ICU admission, mortality, hospital stay). Also consider adjusting for potential confounders such as age, BMI, CRP, and infection severity.

2. Report effect sizes and 95% confidence intervals, not only p-values.

3. The study concludes that dysglycaemia was common but not associated with clinical outcomes. Given the small number of adverse events, this result may be due to limited power rather than a true absence of association. The authors should clearly state the low event rate as a limitation.

4. Discuss whether dysglycaemia might serve as an early biomarker for clinical deterioration, rather than an independent outcome predictor.

5. The analysis would be more informative if additional variables were included such as type and site of infection, nutritional intake, IV glucose, and insulin dosing during admission (noting that severity scores like SIRS/qSOFA are already reported at baseline). Without these, the ability to interpret the mechanisms or contributors to dysglycaemia is limited.

6. Expand discussion on why hypoglycaemia and hyperglycaemia patterns differ between groups.

7. Provide a mechanistic rationale connecting infection-related stress responses with CGM-detected glycaemic patterns.

8. Change the conclusion that CGM “may improve outcomes,” which is not supported by the presented data.

9. Define “severe infection” operationally—were diagnoses microbiologically confirmed or purely clinical?

10. Clarify whether HbA1c data were unavailable or simply not collected; this affects interpretation of undiagnosed diabetes.

**Do you want your identity to be public for this peer review?** For information about this choice, including consent withdrawal, please see our Privacy Policy

Reviewer #1: No

Reviewer #2: No

---

## [Author Response · Author response to Decision Letter 1]

13 Jan 2026

All reviewer comments have been addressed in detail in the uploaded 'Response to Reviewers' document.

Revisions to the manuscript are indicated in the marked-up version with tracked changes ('Revised Manuscript with Track Changes') and are also incorporated in the clean version of the manuscript ('Manuscript') without tracked changes.

---

## [Decision Letter · Decision Letter 1]

11 Feb 2026

Glucose dysregulation in hospitalized non-critically ill patients with a suspected infection: a prospective study using continuous glucose monitoring

PONE-D-25-49860R1

Dear Dr. Bakker,

We’re pleased to inform you that your manuscript has been judged scientifically suitable for publication and will be formally accepted for publication once it meets all outstanding technical requirements.

Kind regards,

Hidetaka Hamasaki

Academic Editor

PLOS One

Additional Editor Comments (optional):

Thank you for submitting the revised manuscript.

Reviewers' comments:

Reviewer's Responses to Questions

**Comments to the Author**

Reviewer #1: All comments have been addressed

2. Is the manuscript technically sound, and do the data support the conclusions?

Reviewer #1: Yes

3. Has the statistical analysis been performed appropriately and rigorously?

Reviewer #1: Yes

4. Have the authors made all data underlying the findings in their manuscript fully available?

Reviewer #1: Yes

5. Is the manuscript presented in an intelligible fashion and written in standard English?

Reviewer #1: Yes

Reviewer #1: (No Response)

**Do you want your identity to be public for this peer review?** For information about this choice, including consent withdrawal, please see our Privacy Policy

Reviewer #1: No

---

## [Editor Report · Acceptance letter]

PONE-D-25-49860R1

PLOS One

Dear Dr. Bakker,

I'm pleased to inform you that your manuscript has been deemed suitable for publication in PLOS One. Congratulations! Your manuscript is now being handed over to our production team.

Kind regards,

on behalf of

Dr. Hidetaka Hamasaki

Academic Editor

PLOS One